# Probiotics, Prebiotics and Epithelial Tight Junctions: A Promising Approach to Modulate Intestinal Barrier Function

**DOI:** 10.3390/ijms22136729

**Published:** 2021-06-23

**Authors:** Elizabeth C. Rose, Jack Odle, Anthony T. Blikslager, Amanda L. Ziegler

**Affiliations:** 1Department of Clinical Sciences, College of Veterinary Medicine, North Carolina State University, Raleigh, NC 27606, USA; ecrose2@ncsu.edu (E.C.R.); atbliksl@ncsu.edu (A.T.B.); 2Laboratory of Developmental Nutrition, Department of Animal Science, College of Agriculture and Life Sciences, North Carolina State University, Raleigh, NC 27607, USA; jodle@ncsu.edu

**Keywords:** probiotics, prebiotics, bioactive compounds, intestinal barrier function, tight junctions, toll-like receptors, intestinal microbiota

## Abstract

Disruptions in the intestinal epithelial barrier can result in devastating consequences and a multitude of disease syndromes, particularly among preterm neonates. The association between barrier dysfunction and intestinal dysbiosis suggests that the intestinal barrier function is interactive with specific gut commensals and pathogenic microbes. In vitro and in vivo studies demonstrate that probiotic supplementation promotes significant upregulation and relocalization of interepithelial tight junction proteins, which form the microscopic scaffolds of the intestinal barrier. Probiotics facilitate some of these effects through the ligand-mediated stimulation of several toll-like receptors that are expressed by the intestinal epithelium. In particular, bacterial-mediated stimulation of toll-like receptor-2 modulates the expression and localization of specific protein constituents of intestinal tight junctions. Given that ingested prebiotics are robust modulators of the intestinal microbiota, prebiotic supplementation has been similarly investigated as a potential, indirect mechanism of barrier preservation. Emerging evidence suggests that prebiotics may additionally exert a direct effect on intestinal barrier function through mechanisms independent of the gut microbiota. In this review, we summarize current views on the effects of pro- and prebiotics on the intestinal epithelial barrier as well as on non-epithelial cell barrier constituents, such as the enteric glial cell network. Through continued investigation of these bioactive compounds, we can maximize their therapeutic potential for preventing and treating gastrointestinal diseases associated with impaired intestinal barrier function and dysbiosis.

## 1. The Emergence of Pre- and Probiotic Therapy

The first campaign for host-friendly bacteria occurred in the late 19th century when Dr. Élie Metchinkoff, who later received the Nobel Prize in Physiology or Medicine, began to prescribe a new treatment for old age: soured milk [1]. Metchinkoff’s rationale for the innovative medicine stemmed from his realization that rural, Bulgarian peasants were outliving their rich, city-dwelling peers. Metchinkoff attributed this lifespan discrepancy to the peasants’ consumption of soured milk, which likely resembled an early-day counterpart to today’s ever-expanding yogurt aisle. Although host-bacterial mutualism is now a well-established paradigm, the mechanisms through which commensal organisms support systemic health are largely unresolved. Given that the intestinal tract hosts the body’s largest bacterial ecosystem, elucidation of this mutualistic relationship relies upon an advanced appreciation of the microbiota’s effects on intestinal homeostasis.

Intestinal homeostasis is largely dependent on careful maintenance of the intestinal barrier, a physical boundary which separates the host from noxious luminal microbes and compounds. Intestinal barrier function is regulated through complex interactions among several intrinsic and extrinsic components, including intestinal epithelial cells (IECs), a superficial mucus layer, the enteric nervous system (ENS) and various populations of immune cells. The maintenance of intact barrier function is largely dependent upon paracellular tight junctions (TJs), which link adjacent IECs at their apical surface and create a polarized monolayer with an apical and basolateral domain. Relocalization and altered expression of TJ protein constituents can result in rapid deterioration of intestinal barrier function, ultimately hindering normal absorptive and secretory activity as well as dissolving the host’s physical barricade against noxious luminal stimuli.

Interestingly, studies utilizing a pig model for intestinal ischemia have demonstrated that barrier recovery of the small intestines is significantly reduced in neonates as compared to juveniles [2]. These findings suggest that infants may be particularly predisposed to serious complications of barrier insult because of their inability to mount a robust reparative response. Neonatal barrier function can be challenged by the birthing process alone, which can prompt adverse, perinatal outcomes such as necrotizing enterocolitis (NEC), a life-threatening condition characterized by inflammation and inappropriate bacterial overgrowth throughout the small and large intestines [3,4,5]. Although most publications demonstrate that probiotic administration for intestinal disease prevention or treatment is overall safe and variably effective, the mechanisms of action remain poorly understood.

The rising popularity of probiotic supplementation has fueled interest in the precise mechanisms through which IECs recognize probiotics, commensal organisms and other microbes. Although this concept was initially dichotomizing, there is now a general consensus that IECs express certain pattern-recognition receptors (PRRs), which are encoded proteins that recognize and bind specific microbial ligands. Although many sub-families of PRRs have been described, a complete review of these subfamilies is beyond the scope of this review. Instead, we will focus on the specific sub-family of toll-like receptors (TLRs) due to emerging evidence for TLR’s interactions with the intestinal microbiota and epithelial cell TJs. Given their direct interaction with intestinal TJs, TLRs are an exciting potential target for probiotic therapy.

Furthermore, we introduce the budding realm of prebiotic supplementation and its effects, both direct and indirect, on the intestinal barrier. Although specific investigations into the effects of prebiotic supplementation on intestinal barrier function are relatively recent compared to those with probiotics, the limited data that have emerged informs exciting prospects for future endeavors. Given their speculated shared health benefits, pre- and probiotics have become increasingly lumped together as bioactive compounds, which are extra-nutritional food constituents that provide health benefits beyond a basic nutritional value. This review aims to highlight how the intestinal TJs, and consequently intestinal barrier function, can be directly and indirectly influenced by specific interactions among gut commensals, TLRs, and select bioactive compounds, namely pre- and probiotics.

## 2. Extrinsic and Intrinsic Perinatal Variables Influence the Neonatal Intestinal Microbiota

Concurrent with probiotics’ rising popularity, manuscripts reporting the genesis of the intestinal microbiota have numerically increased by more than one hundred-fold over the past few decades. Intestinal gut colonization is now speculated to commence in utero, thus challenging the long-standing dogma that the microbiota is established at birth [6,7,8,9,10]. Sequencing technologies have revealed that the neonatal intestine is rapidly populated by microorganisms that, after just a couple of days, exponentially expand to a number exceeding that of all host cells [11]. Initial colonization by aerobic bacteria transforms the intestinal lumen into an oxygen-poor niche that is subsequently dominated by anaerobes [12,13,14]. Many of these anaerobic bacteria are genetically adapted to obtain energy from host-derived glycans including human milk oligosaccharide, a naturally occurring prebiotic present in the diet of breastfed infants [15].

Initial microbial colonization of the neonatal intestine can significantly vary depending on an assortment of factors including delivery method, feeding source, antibiotic administration, and gestational age [16,17,18]. Preterm infants, for instance, often experience sterile formula feedings, widespread use of antibiotics and reduced exposure to maternal microflora. Furthermore, formula fortification with pre- and probiotics has become a high priority given that bovine milk, from which most commercial formulas are derived, is comparatively low in prebiotic oligosaccharides utilized by gut commensals as a primary energy source [19]. Therefore, although developed to protect against pathogen transmission and infection, these protocols diminish early-life exposure to commensal bacteria and can impede the efficiency of gut colonization. Importantly, preterm infants with altered intestinal flora due to antibiotic use are more likely to develop NEC [20]. Necrotizing enterocolitis is among the most common gastrointestinal emergencies in preterm neonates worldwide, occurring in approximately 7% of very low birth weight infants and resulting in mortality rates of up to 30% [3,21,22]. The precise pathogenesis of NEC are unclear but it is presumably multifactorial with several identified risk factors in addition to preterm birth. This knowledge gap in disease pathogenesis, as well as NEC’s alarming occurrence rate, have motivated researchers across the globe to determine whether probiotics may ameliorate or reduce the incidence of NEC. In 2010, meta-analysis of eleven randomized, controlled trials confirmed that preterm, very low birth weight infants who receive probiotic supplementation have a significantly lower risk for developing NEC [23]. A more recent trial demonstrated reduced NEC incidence and severity in infants given a probiotic that contains 4 species of *Bifidobacterium* [24]. These trials provide promise for probiotics’ beneficial effects in neonates, particularly preterm infants.

## 3. IECs Recognize Intestinal Microbes through Toll-Like Receptors

Toll-like receptors contribute to IEC homeostasis by governing appropriate host responses to the intestinal microbes. Downstream effects of nearly all TLRs are mediated by Myd88, which is a cytosolic adaptor protein that converts extracellular signals to intracellular activity including NF-*κ*B activation and cytokine production. A few seminal publications on TLR expression by IECs will be summarized below. For a more detailed discussion, readers are referred to relevant reviews [25,26].

To further investigate the significance of TLR stimulation in barrier function, it is initially important to identify when and where IECs express specific receptors. Particular attention has been paid to TLR2 and TLR4, which recognize pathogen-associated molecule patterns (PAMPs) expressed by gram-positive and gram-negative bacteria, respectively. In mice, TLR expression significantly varies throughout the length of the intestinal tract. Colonic IECs exhibit robust expression of TLR2 and TLR4 while small intestinal IECs express very low levels of each receptor [27]. Spatial compartmentalization of these two receptors is perplexing given that bacterial constituents can be isolated along the entire gastrointestinal tract. Speculatively, it is possible that the small intestinal barrier may be particularly sensitive to injury following robust TLR-signaling. Therefore, decreased TLR expression may mitigate small intestinal injury through reduced bacterial-TLR signaling. Limited TLR expression may also be explained by the fact that, in health, the small intestine contains a sparce microflora compared to the colon. Therefore, small intestinal cells may restrict TLR expression so that they can dedicate energy to more essential functions, namely nutrient absorption.

Where expression is detectable, TLR2 and TLR4 are constitutively localized to the epithelium’s apical surface [28,29,30]. Localization to the apical surface of IECs promotes TLR stimulation by the contents of the intestinal lumen, including commensal bacteria as well as injurious pathogens. Interestingly, TLR expression in undifferentiated, or immature, intestinal epithelial cell lines is limited to the cytoplasmic compartment [30]. Therefore, IECs’ ability to sample the intestinal lumen may evolve with age, possibly paralleling shifts in the microbiota that naturally occur throughout life.

Caution must be taken when proceeding with this singular interpretation, however, as undifferentiated intestinal cell lines are defined by a lack of polarity. It therefore remains plausible that undifferentiated, unpolarized IECs retain TLR domains within their cytoplasmic compartments until distinct apical and basolateral surfaces are defined. This speculation is furthered by a recent demonstration that the largest amount of lipopolysaccharides (LPS) is taken up from IECs along the villus tip rather than within the crypts [31]. Therefore, it is equally plausible that IECs alter TLR expression as they migrate up the crypts to the tip of the intestinal villi.

Differentiation between the beneficial, or at least benign, microbiota and harmful pathogens is further complicated by the fact that TLR-specific ligands are expressed by both commensal and pathogenic organisms. To mitigate this challenge, IECs rely upon a fundamental principle of bacterial pathogenicity: harmful bacteria are more likely to breach the epithelial barrier. Many pathogenic bacteria rely upon a timely production of certain virulence factors to breach the mucosal surface and invade the subjacent parenchyma. *Salmonella* spp., for example, encode a type III secretion system that allows for direct ‘injection’ of Salmonella invasive proteins (SIPs), which stimulate enterocyte apoptosis, consequently creating a physical defect in the epithelial barrier. Therefore, cellular responses may be customized according to the physical location of the TLR on the cell surface. As an example, toll-like receptor 9 (TLR9) is activated by unmethylated CpG domains in bacterial DNA. Downstream effects of TLR9 stimulation are determined by the origin of the receptor’s ligand. TLR9 activation along the apical surface of IECs inhibits NF-*κ*B activation. Conversely, activation from the basolateral surfaces promotes canonical activation and nuclear translocation of NF-*κ*B [32]. Therefore, pathogenic organisms which translocate across the IEC barrier and access basolateral TLR9 are more likely to exert a proinflammatory effect which is intended to clear the pathogens but has the potential to become deleterious causing local tissue destruction and further barrier injury.

Similarly, cellular responses can be customized by restricting the expression of certain TLRs to specific IEC domains. For instance, toll-like receptor 5 (TLR5), which recognizes bacterial flagellin, is expressed along only the basolateral surface of colonic IECs in vitro [33]. Interestingly, TLR5 expression by murine small intestinal epithelial cells gradually decreases throughout the neonatal period until expression is largely restricted to Paneth cells in adult mice [27]. These findings reinforce the speculation that TLR expression may evolve with age. Robust TLR5 expression in neonates may act to provide further protection from pathogenic bacteria while the immature intestinal tract and immune system continue to develop. Alternatively, one may speculate that decreased TLR5 expression by neonatal Paneth cells may predispose the neonate to prolonged and more severe intestinal injury. Paneth cells support the intestinal stem cell niche and are arguably necessary for stem cell proliferation and differentiation [34]. Therefore, decreased TLR5-mediated stimulation by Paneth cells may result in hindered epithelial proliferation and recovery of the neonatal intestines following injury. By promoting a healthy commensal microbial community with pre- and probiotic interventions, these pathogens can be reduced by competitive inhibition, quite possibly preventing their harmful TLR-mediated effects on the intestinal barrier.

## 4. Intestinal Tight Junctions Are Regulated by Bacterial Stimulation of Toll-Like Receptors

Intestinal TJs regulate epithelial permeability through the dynamic nature of their three-dimensional, multiprotein conformation. Essential constituents of all TJs include integral membrane proteins, such as occludin and various claudins, as well as cytoplasmic plaque proteins, such as zonular occludins-1 (ZO-1) and zonular occludins-2 (ZO-2). Previous reviews have detailed these constituent’s differing effects on intestinal permeability [35,36,37]. Provided that IECs express specific TLRs along their apical and basolateral domains, researchers have reasonably speculated that microbial ligands regulate TJ proteins through their interactions with TLRs. Consequently, there is a steady increase in the number of studies that investigate alterations in specific TJ proteins following supplementation with probiotics. For example, activation of MyD88, the cytosolic adaptor protein for most TLRs, is required for appropriate claudin-3 expression in neonatal mice. MyD88^−/−^ neonatal mice demonstrate impaired intestinal barrier function with increased permeability to large polysaccharides. Therefore, TLR-mediated induction of claudin-3 protein expression may be necessary for appropriate TJ formation and maintenance [38]. Furthermore, maturation of barrier function and claudin-3 protein expression in mouse pups is induced by enteric feeding of *Lactobacillus rhamnosus* GG, which is an intestinal commensal species and a commonly used probiotic [38]. Taken together, these findings suggests that expression of crucial barrier-forming TJ proteins, such as claudin-3, may be induced via TLR signaling associated with select probiotics.

Specific TLRs play distinct roles in the regulation of intestinal TJs. Following treatment with a TLR-2 ligand, for instance, Caco-2 monolayers demonstrate robust apical redistribution of ZO-1 as well as increased intracellular protein kinase C (PKC) activity [39]. This may suggest that TLR2 stimulation promotes structural modifications in TJs by promoting PKC activity. Therefore, deficient levels of host-protective TLR2 ligands may compromise barrier function. For instance, antimicrobial administration could compromise barrier function by decreasing luminal numbers of gram-positive gut commensals such as *Lactobacillus* spp. and *Bifidobacterium* spp., both of which are important sources of TLR2-stimulating ligands [40].

Notably, TLR activation does not always result in increased barrier function. Stimulation of TLR4 by LPS causes decreased barrier function in mouse models and human intestinal epithelial cell lines [41]. This detrimental effect is attributable to TLR4/MyD88 upregulation of myosin light chain kinase (MLCK). Myosin light chain kinase induces the opening of the TJs through contraction of actin-myosin filaments, followed by internalization of TJ proteins [41,42,43,44,45]. Therefore, intestinal homeostasis and barrier function requires a delicate balance between commensal- and pathogen-induced activation of TLRs.

To maintain this delicate balance and further investigate the effects of deliberate TLR stimulation, researchers must complement their research studies with a foundational understanding of TLR pathophysiology. Each TLR is defined by a unique combination of stimulating ligands, co-receptors, adaptor proteins and effector pathways. Given TLR-2′s potential as a therapeutic target for intestinal barrier function, its unique protein dynamics merit careful consideration. Interestingly, TLR-2 is the only TLR described thus far to form functional heterodimers with two different types of TLR as well as a large number of non-TLR molecules [46,47,48,49,50]. This unique ability to interact with multiple co-receptors and adaptor proteins, such as Declin-1 and CD14, facilitates TLR-2′s ability to bind a broad repertoire of structurally diverse ligands [51,52,53]. A thorough understanding of TLR-2′s various co-receptors, adaptor proteins and ligands can inform probiotic formulations so as to effectively and specifically stimulate TLR-2 and promote it’s beneficial effects on intestinal TJs.

## 5. Probiotics Promote Barrier Function through Regulation of Tight Junction Proteins

*Lactobacillus* and *Bifidobacterium* species are prominent genera within the human gut microbiota and have become the most commonly-used probiotics in human medicine, including obstetrics [54,55]. The proposed mechanism of these commensals was initially based upon in vitro experiments that utilize intestinal epithelial cell lines. Preincubation of Caco-2 cells with specific strains of *Lactobacillus plantarum* and *Lactobacillus rhamnosus* attenuate pathogen-mediated disruptions in the intestinal epithelial barrier, including a highly reductionist bacterial LPS in vitro model of NEC, which entailed TJ disruption through LPS derived from *Escherichia coli* [56,57,58]. Although NEC is often associated with bacterial overgrowth and dysbiosis, the exact pathogenesis remains unknown and is generally accepted to be multifactorial. Therefore, an LPS model is likely an over-simplification of NEC and until the pathogenesis of NEC is further elucidated, conclusions gleaned from such simplified models should be made with caution. However, this probiotic-mediated barrier protection in a model of bacterial intestinal insult is certainly worth noting.

Certain *Lactobacillus* species abate barrier disruption through upregulation of TJ proteins. *L. acidophilus* and *L. plantarum* increase occludin protein expression within in vivo and in vitro models, respectively [59,60]. In addition to increasing occludin protein expression, *L. plantarum* induces apical relocalization of ZO-1 and occludin through stimulation of TLR2 [58,61]. However, Caco-2 cell coincubation with *L. plantarum* simultaneously results in increased transcription of genes involved in tight junction disassembly and occludin degradation [60]. Therefore, increased occludin expression and apical localization may actually be a protective response provoked by initial bacterial-mediated degradation, not maintenance, of the TJs.

While *Bifidobacterium* supplementation has been repeatedly shown to decrease the incidence and severity of NEC, the potential mechanism by which this gut commensal exerts its beneficial effects has only recently been described [62,63,64]. Upon the onset of NEC, internalization of claudin 4 and occludin disrupts the intestinal TJs [64]. Oral supplementation with *B. infantis* prevents this disturbance by preserving claudin 4 and occludin localization along the vicinity of the TJs. Therefore, *Bifidobacterium* supplementation preserves intestinal barrier function, and consequently prevents severe NEC, though the maintenance of TJ conformation.

*Bifidobacterium* and *Lactobacillus* spp. are the primary constituents of a commercial probiotic formulation called VSL#3. For nearly 20 years, VSL#3 has dominated the dietary supplements market by claiming to mitigate clinical symptoms of certain gastrointestinal disorders. The formulation’s beneficial effects are hypothesized to result from bacterial-mediated preservation of TJ protein expression and localization as well as inhibition of intestinal epithelial cell apoptosis. Increased colonic permeability and epithelial cell apoptosis resulting from DSS-induced colitis in mice were prevented by VSL#3 supplementation, which additionally increased expression of ZO-2, occludin and several claudins [65]. However, the VSL#3 dose required to produce these beneficial effects in this mouse colitis model is twice that recommended for inflammatory bowel disease treatment in humans. These findings warrant a reevaluation of current probiotic dose recommendations for VSL#3 as well as other commercial probiotics. Probiotics advertised as dietary supplements do not require approval by the United States Food and Drug Association before marketing. Therefore, many probiotic manufacturers have not had the incentive to research physiologic differences among various dosages. As the probiotics market continues to expand, the supplementation efficacy and dosing demands more intensive study.

For some probiotics, realization of their beneficial effects relies upon prior disruption of TJ homeostasis. For instance, *E. coli Nissle 1917* (EcN) is a gram-negative probiotic that has been studied since the discovery of its potential benefit during World War I. Alfred Nissle isolated the strain in 1917 from the feces of a German soldier after noting that the soldier did not develop diarrhea despite a regional Shigellosis outbreak [66]. Nissle speculated that gastrointestinal EcN populations prevented the soldier from falling ill. Nearly one century following this initial isolation, studies have expanded upon Nissle’s findings by noting that incubation of T84 human intestinal epithelial cells with EcN alone results in no significant changes to intestinal epithelial barrier function [67]. However, following T84 co-incubation with enteropathogenic *E. coli* (EPEC) and associated barrier disruptions, EcN supplementation restores barrier permeability to nearly the same value as seen in control cells. The beneficial properties of EcN might therefore be realized post-infection and may not require preventative supplementation, which is clinically challenging and often not feasible. Additionally, in concurrence with decreased barrier permeability, EcN supplementation results in increased ZO-2 expression and robust ZO-2 relocalization to the TJs [67]. Although these early findings do not prove absolute causation, they suggest that EcN’s beneficial effects on intestinal barrier function may be modulated by the bacteria’s regulation of ZO-2 expression and localization.

*E. coli Nissle 1917* supplementation can enhance IEC expression of ZO-1 under healthy and inflammatory conditions in vivo. However, EcN administration following DSS-induced colitis does not significantly decrease colonic permeability [68]. Therefore, the practical significance of increased ZO-1 expression in vivo remains equivocal. Interestingly, ZO-2 expression is not altered upon in vivo administration of EcN, which is unexpected given that it has been shown to increase in vitro [67]. One potential explanation for this discrepancy in ZO-2 expression is that the aforementioned in vitro studies utilized a human intestinal epithelial cell line while the in vivo studies used a mouse model. The pathophysiology of mouse intestinal TJs may differ from that of human intestinal TJs. Although animal models play an invaluable role in the advancement of human medicine, species variation can limit data extrapolation from these models. Future studies evaluating EcN effects on TJ protein expression across numerous species, including humans, under normal conditions as well as models of intestinal injury would provide interesting insight into how mechanisms of intestinal repair may vary across species.

The skin and gut barrier share many morphological and pathophysiological similarities. Therefore, some researchers have utilized normal human epidermal keratinocytes (NHEK) to infer potential mechanisms by which probiotics modulate intestinal TJs. Keratinocyte cell cultures demonstrate increased barrier function and TJ protein expression following treatment with bacterial lysates from certain strains of *Lactobacillus* and *Bifidobacterium* spp. [69]. These phenotypes are abolished with neutralization of TLR2, suggesting that these bacteria augment TJ closure through TLR2 stimulation. Unfortunately, key physiologic differences between keratinocytes and enterocytes, notably the robust secretory and absorptive capacity of the latter, prohibit a blind application of these findings to the intestinal barrier. Instead, it would be interesting to repeat the referenced experiment using intestinal cell lines or in vivo models. A summary of selected probiotics’ target TLRs and proposed beneficial effects on intestinal tight junctions is available in Table 1.

## 6. Specific Bacterial Components May Contribute to Probiotics’ Beneficial Effects

Given that different probiotic formulations exhibit similar effects on the intestinal epithelial barrier, we can plausibly speculate that probiotics’ mechanisms of action are regulated by shared bacterial components rather than specific bacterial species. In other words, probiotics’ effects on intestinal TJs and barrier function may be mediated by distinct structural and chemical bacterial constituents rather than the bacterium as a whole.

Soluble proteins isolated from *Lactobacillus rhamnosus* ameliorate intestinal epithelial barrier disruption in vitro by preventing injury-induced redistribution of occludin and ZO-1 [71]. Furthermore, pretreatment with the isolated proteins prevents increased barrier permeability to ions and large sugar molecules. The introduction of a PKC- or MAP-kinase inhibitor, however, attenuates these phenotypes. Therefore, soluble proteins secreted by select probiotic bacteria exert their protective effects through a PKC- and MAP-kinase-dependent mechanism.

Indole is an intercellular signaling molecule produced by many gram-positive and -negative organisms. Repeated indole administration results in increased protein expression of vital TJ constituents as well as resistance to DSS-induced mucosal damage and colitis [72]. Therefore, the protective effects of some probiotics may be reliant upon indole production and signaling.

Peptidoglycans are the primary constituent of most bacterial cell walls. Co-incubation of NHEKs with bacterial peptidoglycans results in increased barrier function [69]. Again, relying upon the morphologic similarities between the dermal and intestinal epithelium, these results suggest that bacterial cell walls contribute to probiotics’ promotion of intestinal homeostasis and barrier function. However, NHEK co-incubation with bacterial peptidoglycans does not alter TJ protein expression levels. Therefore, peptidoglycans appear to promote barrier function through mechanisms that do not influence TJ protein expression levels. These findings force us to acknowledge that protein activity is not determined by expression levels alone. Peptidoglycan and other bacterial ligands may influence TJ proteins through structural modifications, such as phosphorylation or even ubiquitination. Investigations into probiotics’ mechanisms of action must therefore expand beyond simple protein expression and localization to include post-translational modifications.

Investigations into the role of peptidoglycans in intestinal homeostasis is further complicated by the ligand’s nearly ubiquitous nature: peptidoglycan is present in the cell wall of both commensal and pathogenic bacteria. Therefore, there must be some physical or chemical property of specific peptidoglycans that differentiate these two populations of bacteria. In fruit flies, immune cells rely on peptidoglycan recognition proteins (PGRP), which are a type of PRR similar to TLRs, to differentiate bacterial peptidoglycans [73]. More specifically, PGRPs stimulate customized cellular responses based upon the amino acid that resides in specific positions of the peptidoglycan peptide [74,75]. The binding of some peptidoglycans, and consequently some bacteria, illicit a potent immune response while other bacteria and their associated peptidoglycan will dampen the immune response.

Although these findings are not directly translatable to the mammalian intestinal microbiota, they illuminate that TLRs are only one of several types of PRRs. Other PRRs, including PGRP, have been implicated in the homeostasis of the gut barrier as well as the maintenance of gut commensal populations in mice [73,76]. Therefore, PRRs other than TLRs are promising therapeutic targets for repairing intestinal epithelial TJs and promoting intestinal barrier function.

## 7. Prebiotics Promote Barrier Function through Microbial-Dependent and -Independent Mechanisms

Despite probiotics’ promising effects on intestinal barrier homeostasis and repair, potential drawbacks and limitations of probiotic therapy cannot be ignored. Over the past few decades, reviews have highlighted potential dangers of probiotic therapy in perinatal infants, immunocompromised individuals and patients with compromised intestinal barrier function [77,78,79]. The hindered immune system and barrier function in these demographics exponentially increase the risk for microbes, even those considered to be commensal, to breach the mucosal barrier and establish multisystemic infections. For instance, a recent report described three unrelated cases of *Bifidobacterium longum* bacteremia in very low birth weight infants following administration of a commercially available probiotic preparation [80].

This and similar case series emphasize that very few studies have investigated appropriate doses for probiotic therapy. To date, essentially every study has utilized not only a different probiotic preparation but also a different dosage regime. Investigating the proper doses for safe and effective probiotic supplementation is further complicated by the fact that probiotics must endure the hostile journey from the oral cavity to the intestines as well as survive within the intestinal lumen long enough to illicit their beneficial properties.

Limitations of probiotic therapy have organically fueled an emergence of prebiotic supplementation and research. Prebiotics directly influence the composition of the gut microbiota and stimulate growth of beneficiary commensals such as *Bifidobacterium* spp. and *Lactobacillus* spp. [81,82,83,84,85]. The term “prebiotics” was introduced to the scientific community in a 1995 publication that defined the bioactive substances as “nondigestible food ingredients that beneficially affect the host by selectively stimulating the growth and/or activity of one or a limited number of bacterial species already resident in the colon.” [86] Since this publication, prebiotics’ proposed benefits have expanded beyond colonic health to include the treatment and prevention of select gastrointestinal, skin, genitourinary and skeletal diseases in both humans and animals [81,83,87,88,89,90,91,92,93,94,95]. For this reason, the definition of prebiotics was updated in 2017 to encompass substrates that are “utilized by host microorganisms conferring a health benefit.” [87]

Given probiotics’ beneficial effects on the intestinal barrier, it is not surprising that prebiotic supplementation similarly promotes intestinal barrier function and repair [84,96,97,98,99,100]. In vitro and in vivo studies are largely centered around select formulations of numerous oligosaccharides and polysaccharides, both of which have gradually come to dominate the forefront of prebiotic research [38,83,90,97,99,101,102]. Many publications highlight that not all formulations can be treated equally because the degree of beneficial effects is unique to each tested prebiotic as well as each prebiotic–probiotic combination. These findings introduce the complexity of synbiotics, which rely upon synergism between prebiotics and probiotics to amalgamate the two bioactive substances into a single supplement.

Mimicking probiotics’ proposed mechanisms of action, many prebiotics promote intestinal barrier function [85] through modulation of intestinal TJs. In vitro supplementation with inulin fermentation products results in the significant upregulation of TJ genes including occludin, claudin-3 and ZO-1 [102]. Supplementation with fructooligosaccharides and butyrate, which is a bacterial metabolite produced during prebiotic fermentation, results in the redistribution of select proteins, including ZO-1 and occludin, to the vicinity of the TJs [103,104]. In vivo and in vivo studies demonstrated that galactooligosaccharide pretreatment results in upregulation of ZO-1, occludin and claudin-1 gene expression in LPS-challenged mice [105,106]. A key feature of this particular study is prebiotic administration prior to intestinal injury. As previously mentioned, prebiotic and probiotic pretreatment is often unrealistic in human medicine given that many intestinal injuries are unpredictable. In the case of preterm infants with NEC, pretreatment with bioactive compounds is simply impossible. Therefore, future endeavors must demonstrate beneficial effects of pre- and probiotic supplementation following, not just prior, to intestinal injury in order to support impactful advancements in clinical medicine.

Prebiotics’ direct effects on the gut microbiota is a reasonable explanation for these observed changes in TJ protein expression and distribution. Given that prebiotics stimulate the growth of select probiotic species, it is unsurprising that prebiotic supplementation mirrors the beneficial effects produced by the same probiotics that those carbohydrates fuel. Furthermore, prebiotic supplementation results in robust activation of AMP-activated protein kinase (AMPK) in conjunction with the described changes to the epithelial barrier [98,103,104,106]. Prebiotic-mediated activation of AMPK may therefore play a significant role in prebiotics’, and potentially probiotics’, effects on intestinal tight junctions. Contrarily, data from two of the aforementioned studies suggest that some of prebiotics’ beneficial effects may occur independently of the surrounding microbiota [96,104]. Although these microbial-independent effects are less robust than those produced with concurrent probiotic supplementation, the remarkable possibility of prebiotic-mediated effects despite concurrent dysbiosis, is undoubtedly worth pursing further. A summary of selected prebiotics’ target TLRs and proposed beneficial effects on intestinal tight junctions is available in Table 2.

## 8. Future Directions: Embracing the Enteric Glial Cell Network, Large Animal Models, and Promising Clinical Interventions

As previously mentioned, the epithelial mucosa and its associated tight junctions are only one piece to the intricate puzzle that constitutes the intestinal barrier. Intestinal barrier function is additionally modulated by the ENS, which is a complex network of densely intertwined neurons and glial cells that extends along the entire length of the gastrointestinal tract. Enteric neurons regulate intestinal barrier function through a variety a mechanisms including the promotion of epithelial cell turnover and increased expression of TJ proteins [107,108,109]. Given that enteric neurons express select TLRs, their regulatory effects are hypothesized to be mediated, at least in part, by their interactions with the intestinal microbiota [110,111,112]. However, an exciting frontier of intestinal barrier research, a comprehensive review of enteric neuron’s regulation of the intestinal barrier function and TLR expression is beyond the scope of this review and readers are referred to relevant publications.

While the neuro-epithelial unit has received extensive evaluation in intestinal barrier research, enteric glial cells have been historically characterized by their supportive role in neuronal maintenance and activity. However, emerging research demonstrates that the enteric glial cell (EGC) network acts as more than a mere support system to the enteric neurons and actually plays a direct role in intestinal barrier function. The EGC network is consequently gaining increased interest in the gastrointestinal community due to its implication in intestinal barrier homeostasis and repair, including regulation of TJs and cell motility [107,113,114,115,116,117].

Interestingly, microbial fermentation products, specifically short-chain fatty acids (SCFA), have been shown to modulate EGC network development [118]. The presence of the intestinal microbiota additionally contributes to appropriate, perinatal development of EGCs [119,120]. The microbiota’s effects on the EGC network development are postulated to be regulated, at least in part, by microbes’ direct interactions with intestinal neurons and glia. Enteric neurons and glia within the myenteric and submucosal plexi of adult rodents express TLRs 2, 3, 4, 7 and 9 [111,112]. Given that enteric neurons and glia have no known direct contact with the intestinal lumen, TLR-activation by intestinal microbes is presumably limited to the subepithelial compartment. Therefore, TLR-mediated stimulation of the EGC must occur consequent to barrier disruption by pathogenic organisms or, in physiologic conditions, translocation of gut commensals across the mucosal barrier. Furthermore, TLR protein expression by the EGC network fluctuates depending on the specific microbial organisms that are present. For instance, TLR2 protein expression is upregulated when EGCs are co-cultured with a specific species of *Lactobacillus* [121]. Given EGC expression of certain TLRs and the networks’ regulatory effects on barrier function, we can reasonably speculate that pro- and prebiotics’ preservation of intestinal barrier function may be mediated, or at least enhanced, through their interactions with the EGC network. Figure 1 illustrates our current understanding of TLR expression of IECs and the EGC network.

In order to unravel this expanding web of intestinal barrier regulation, future endeavors must further investigate these interactions among the intestinal microbiome, EGC and nutrition. As a natural source of SCFAs, oligosaccharides are a rational stepping stone for these future endeavors. Oligosaccharide supplementation has already shown to increase intestinal epithelial transepithelial resistance, a marker of intestinal barrier function, as well as ileal lactobacilli concentrations in formula-fed pigs [122,123]. Insights into prebiotics’ fermentation profiles and synergistic activity with specific probiotics may provoke even more interesting results, eventually informing future symbiotic supplementation formulas and protocols for mitigating gastrointestinal diseases.

These promising studies on oligosaccharide supplementation introduce another exciting venture in gastrointestinal research: porcine models of human disease. A recent review by Ziegler et al. highlights fundamental physiologic, anatomic and nutritional similarities between pigs and humans [124]. Notably, pigs and humans share similar gut microbial profiles while the murine microbiota is significantly divergent [125,126,127]. Taking these interspecies comparisons into account, pigs can undoubtedly act as powerful models for upcoming investigations into intestinal barrier function and its relationship with the microbiome, nutritional supplements and the EGC [128,129]. In vivo studies in suckling piglets have already demonstrated that early-life supplementation with galactooligosaccharides results in significantly increased ZO-1 and claudin-1 protein expression [106]. These studies did not directly evaluate intestinal barrier function, however, therefore the significance of these results with respect to upholding barrier function requires further investigation.

With the introduction of germ-free and gnotobiotic pigs, researchers can further embrace the porcine model by investigating how the gestational maternal microbiota affects neonatal health. Chapter 2 of this review highlights emerging evidence that the neonatal intestinal microbiota begins to develop in utero. Therefore, the composition of the maternal microbiota throughout gestation may directly influence that of the developing neonate and consequently influence the pathophysiology of the neonate’s intestinal barrier. Given that the intestinal microbiota of pigs and humans is remarkably similar, porcine models may be utilized to investigate whether the maternal intestinal microbiota can be optimized through pre- and probiotic supplementation throughout gestation. Furthermore, such models can investigate whether these supplementation protocols are significantly correlated to the composition of the neonatal intestinal microbiota as well as neonatal intestinal disease outcome.

As many researchers have focused on the effects of pre- and probiotics on intestinal barrier function, others have investigated pharmaceuticals that directly target intestinal TJs. For instance, Larazotide acetate is a small, TJ-directed drug that is currently in clinical trials for treatment of celiac disease, the pathophysiology of which includes disruptions to the intestinal TJs [130,131,132,133]. Several in vivo and in vitro experiments have demonstrated that Larazotide administration prevents relocalization of crucial TJ-proteins, ultimately resulting in quantifiably enhanced epithelial barrier integrity [134,135,136]. Taken together, these studies introduce the exciting prospect of compounding pre- and probiotic’s beneficial effects with Larazotide or similar TJ-protein-directed pharmaceuticals. Future studies should investigate the therapeutic potential of timed treatment protocols in which initial administration of TJ-protein-directed pharmaceuticals enhances TJ integrity to prevent continued bacterial stimulation and subsequent pre- and probiotic supplementation carefully restores a balanced intestinal microbiota.

## 9. Conclusions

Probiotics’ beneficial effects on intestinal barrier function are at least partially mediated through organismal stimulation of TLRs, particularly TLR2, and sequential changes to TJ protein expression and localization. Prebiotic supplementation produces similar beneficial effects on barrier function, although their mechanisms of action have yet to be extensively evaluated. These findings implicate probiotic and prebiotic supplementation as a reasonable preventative and treatment modality for disease syndromes defined by increased intestinal permeability. Some of these disease syndromes, including NEC in preterm infants, have already shown a positive response to probiotic therapy. Future endeavors must expand upon these findings to unravel the intestinal barrier’s dynamic interactions with probiotics, prebiotics and enteric glia. Pigs can act as a powerful translational model to enhance these studies and provide innovative techniques for solving unresolved quandaries. Dietary supplementation with bioactive compounds remains an exciting frontier and presents promising possibilities for therapeutic and preventative modalities in gastrointestinal medicine.

## Figures and Tables

**Figure 1 ijms-22-06729-f001:**
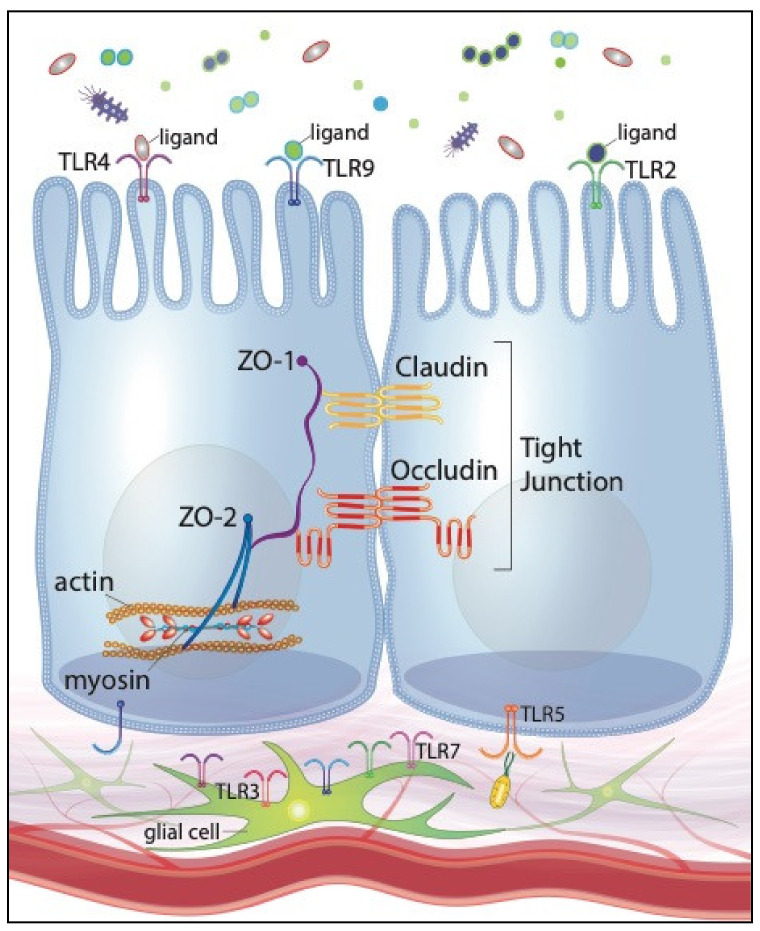
Toll-like receptor expression of intestinal epithelial cells (IECs) and the enteric glial cell (EGC) network, illustrating positive effects on tight junctions and intestinal barrier function.

**Table 1 ijms-22-06729-t001:** Select probiotics target TLRs and proposed beneficial effects on intestinal tight junctions.

Probiotic	Target TLR(s)	Tight Junction Effects	Reference(s)
*Lactobacillus rhamnosus* GG	TLR2, TLR9	Increased claudin-3 protein expression	[38,70]
*Lactobacillus acidophilus*	TLR2	Increased occludin protein expression	[59]
*Lactobacillus plantarum*	TLR2	Increased occludin protein expression and apical redistributionIncreased ZO-1 apical redistribution	[58,60,61]
*Bifidobacterium infantis*	TLR2	Preserves apical distribution of claudin-4 and occludin	[64]
*E. coli Nissle 1917*	TLR4	Increased ZO-2 mRNA and protein expression and apical redistributionIncreased ZO-1 mRNA and protein expression	[67,68]
*Lactobacillus rhamnosus*	TLR2	Preserves apical distribution of ZO-1 and occludin	[71]

**Table 2 ijms-22-06729-t002:** Select prebiotics target TLRs and proposed beneficial effects on intestinal tight junctions.

Prebiotic	Tight Junction Effects	Reference
Inulin	Increased occludin, claudin-3 and ZO-1 RNA expression	[102]
Fructo-oligosaccharide	ZO-1 and occludin apical redistribution	[104]
Galactooligosaccharide	Increased ZO-1, occludin and claudin-1 protein expression	[105,106]

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
