# Peer review of "Probiotics, Prebiotics and Epithelial Tight Junctions: A Promising Approach to Modulate Intestinal Barrier Function"

_ijms, 2021, doi:10.3390/ijms22136729_

Round 1

Reviewer 1 Report

Dear Authors,

This is an excellent review, eloquently written yet concise. I only have some minor suggestions to include / complete the story:

You touch very briefly on the maternal / placental microbiome. Nowadays there are indications that the maternal microbiome can play an important role in neonatal health. I suggest to include a paragraph to discuss the merit of this strategy to give the neonates a good start by optimising the mother's microbiome.

I fully agree with the importance of a solid understanding of TLR dynamics to use as a possible tool for improving gut health. As TLR2 is a 'promiscuous' receptor which dimerizes with a range of different coreceptors (TLR1, TLR6, Dectin, and the list goes on), please include a paragraph where you discuss the dynamics of TLR2 and its co-receptors in more depth, for example by referring a bit more to prof. E. Cario's work and others. For further reading I can also recommend the work by prof. P. de Vos and Dr. L.M. Vogt on the ligand-related mechanisms of inulin-type fructans on immune cells and intestinal barrier function.

With the risk of being out of scope, you could consider mentioning future research into use (or perhaps combined use/timed regimen with pre & probiotics) of pharmaceuticals such as larazotide (used now in celiac disease) to close the intestinal barrier, I think it would fit into the rationale of protecting the basolateral TLRs from further unwanted stimulation, if it is then followed with the next step of carefully populating and (re)balancing the neonatal gut microbiome.

I recommend this manuscript for Acceptance with minor changes.

Author Response

“I suggest to include a paragraph to discuss the merit of [maternal /placental microbiome strategy] to give the neonates a good start by optimising the mother's microbiome. Suggestion to introduce pre-parturition manipulation of maternal microbiome as a means to support neonatal health”

We agree that this is a very interesting and novel approach that merits expanded discussion within this review. An additional paragraph was added to Chapter 7 to discuss the potential for supporting neonatal health through pre-parturition manipulation of the maternal microbiota. This concept was further expanded upon by suggesting experimental methods for researchers to investigate the efficacy of this strategy, including suggested use of porcine models. 

“… please include a paragraph where you discuss the dynamics of TLR2 and its co-receptors in more depth, for example by referring a bit more to prof. E. Cario's work and others. For further reading I can also recommend the work by prof. P. de Vos and Dr. L.M. Vogt on the ligand-related mechanisms of inulin-type fructans on immune cells and intestinal barrier function.

We agree that the complex dynamics of TLR signaling with its co-receptors warrants an additional paragraph for expanded discussion. We have included a concise introduction to TLR2’s unique protein dynamics at the end of Chapter 4 in our revised manuscript. We highlight the significance of these receptor interactions in the context of pre- and probiotic supplementation as well as provide several important references to which the reader can refer for additional information.

“… consider mentioning future research into use (or perhaps combined use/timed regimen with pre & probiotics) of pharmaceuticals such as larazotide (used now in celiac disease) to close the intestinal barrier, I think it would fit into the rationale of protecting the basolateral TLRs from further unwanted stimulation, if it is then followed with the next step of carefully populating and (re)balancing the neonatal gut microbiome.”

We appreciate the clinical importance of discussing co-therapy with bioactive nutrients alongside precise and direct pharmaceutical interventions and we have revised our manuscript to include this concise discussion without overly broadening the scope of our review. A new paragraph at the end of chapter 7 introduces this intriguing suggestion for synchronous or sequential administration of TJ-targeting pharmaceuticals and pre/probiotics. We specifically discuss Larazotide’s mechanism of action and how Larazotide administration may augment the beneficial effects and pre- and probiotic supplementation.

Reviewer 2 Report

This review is generally well written and covers most aspects of the regulatory mechanisms of probiotics and prebiotics on the intestinal epithelial (and non-epithelial) barriers. References are uptodate and figures are of good quality.

However, it is intriguing why the authors did not fully describe possible interactions in relation to the enteric nervous system ? They only focused on glial cells which play supportive role in relation to the enteric neurons. Thus, chapter “7. Future directions: Embracing the enteric nervous system and large animal models” has, in fact, not much common with the role of ENS. Neuronal control is of special importance because the function of epithelial barrier is not only regulated by luminal factors but also by internal (ENS). Moreover, the expression of TLR4 was found on enteric neurons and some neuropeptides played a protective role against LPS-induced neuronal death (see Arciszewski et al. (2008) Vasoactive intestinal peptide rescues cultured rat myenteric neurons from lipopolysaccharide induced cell death. Regulatory Peptides 146(1-3):218-23). Because, this must be included in the text I suggest the authors to partially reedit the manuscript (maybe even add a new chapter).

Author Response

“…it is intriguing why the authors did not fully describe possible interactions in relation to the enteric nervous system ? They only focused on glial cells which play supportive role in relation to the enteric neurons. Thus, chapter “7. Future directions: Embracing the enteric nervous system and large animal models” has, in fact, not much common with the role of ENS. Neuronal control is of special importance because the function of epithelial barrier is not only regulated by luminal factors but also by internal (ENS).”

We appreciate the reviewer’s thoughtful comments and concur that the enteric nervous system (ENS) remains an exciting frontier in intestinal barrier research. We agree that our original title for chapter 7 was overly broad and consequently have revised it to more accurately convey that only the enteric glial cell (EGC) network, not the entire ENS, will be discussed in depth. For this review, we have chosen to highlight the EGC network due to emerging and evolving discoveries of its direct role in intestinal barrier function. As the reviewer mentioned in their comments, historic dogma has labeled the glial cell as a mere support cell to the neuron. Therefore, chapter 7 aims to emphasize current data that strongly supports the EGC as a significant regulator of intestinal barrier function that warrants further investigation. This being said, we are excited by the prospect of writing a follow-up review that provides a comprehensive evaluation of TLRs expressed by the whole ENS, including enteric neurons, and their potential role(s) in intestinal barrier function. For readers who wish to learn more about enteric neurons at this time, we have provided a brief overview of their role in the intestinal barrier function as well as citations for several impactful publications that delve deeper into this topic.

“Moreover, the expression of TLR4 was found on enteric neurons and some neuropeptides played a protective role against LPS-induced neuronal death (see Arciszewski et al. (2008) Vasoactive intestinal peptide rescues cultured rat myenteric neurons from lipopolysaccharide induced cell death. Regulatory Peptides 146(1-3):218-23).”

Thank you for highlighting this important manuscript. We have included a mention of this study and cited this reference in the expanded first two paragraphs of chapter 7, and pointed the readers to more in-depth literature for comprehensive understanding on TLR signaling in enteric neurons.

Round 2

Reviewer 2 Report

The explanations and corrections presented by the authors are satisfactory.